# Implementation of the advanced HIV disease care package with point-of-care CD4 testing during tuberculosis case finding: A mixed-methods evaluation

Tinne Gils[1,2]*, Mashaete Kamele[3], Thandanani Madonsela[4], Shannon Bosman[4], Thulani Ngubane[4], Philip Joseph[4], Klaus Reither[5,6], Moniek Bresser[5,6], Erika Vlieghe[2], Tom Decroo[1], Irene Ayakaka[3], Lutgarde Lynen[1], Alastair Van Heerden[4,7]

1 Department of Clinical Sciences, Institute of Tropical Medicine, Antwerp, Belgium, 2 Global Health Institute, University of Antwerp, Wilrijk, Belgium, 3 SolidarMed, Partnerships for Health, Butha-Buthe, Lesotho, 4 Centre for Community Based Research, Human Sciences Research Council, Pietermaritzburg, South Africa, 5 Clinical Research Unit, Swiss Tropical and Public Health Institute, Allschwil, Switzerland, 6 Medical Outpatient Department, University of Basel, Basel, Switzerland, 7 MRC/Wits Developmental Pathways for Health Research Unit, University of the Witwatersrand, Johannesburg, South Africa

* tgils@itg.be

## Abstract

During TB-case finding, we assessed the feasibility of implementing the advanced HIV disease (AHD) care package, including VISITECT CD4 Advanced Disease (VISITECT), a semiquantitative test to identify a CD4≤200cells/μl. Adult participants with tuberculosis symptoms, recruited near-facility in Lesotho and South-Africa between 2021–2022, were offered HIV testing (capillary blood), Xpert MTB/RIF and Ultra, and MGIT culture (sputum). People living with HIV (PLHIV) were offered VISITECT (venous blood) and Alere tuberculosis-lipoarabinomannan (AlereLAM, urine) testing. AHD was defined as a CD4≤200cells/μl on VISITECT or a positive tuberculosis test. A CD4≤200cells/μl on VISITECT triggered Immy cryptococcal antigen (Immy CrAg, plasma) testing. Participants were referred with test results. To evaluate feasibility, we assessed i) acceptability and ii) intervention delivery of point-of-care diagnostics among study staff using questionnaires and group discussions, iii) process compliance, and iv) early effectiveness (12-week survival and treatment status) in PLHIV. Predictors for 12-week survival were assessed with logistic regression. Thematic content analysis and triangulation were performed. Among PLHIV (N = 676, 48.6% of 1392 participants), 7.8% were newly diagnosed, 81.8% on ART, and 10.4% knew their HIV status but were not on ART. Among 676 PLHIV, 41.7% had AHD, 29.9% a CD4≤200cells/μl and 20.6% a tuberculosis diagnosis. Among 200 PLHIV tested with Immy CrAg, 4.0% were positive. The procedures were acceptable for study staff, despite intervention delivery challenges related to supply and the long procedural duration (median: 73 minutes). At 12 weeks, among 276 PLHIV with AHD and 328 without, 3.3% and 0.9% had died, 84.8% and 92.1% were alive and 12.0% and 7.0% had an unknown status, respectively. Neither AHD nor tuberculosis status were associated with survival. Implementing AHD care package diagnostics was feasible during tuberculosis-case finding. AHD was prevalent, and not

**Data Availability Statement:** The data that support the findings of this study are available in figshare at http://doi.org/10.6084/m9.figshare.24434668.

**Funding:** This project is part of the European and Developing Countries Clinical Trials Partnership 2 (EDCTP2) programme supported by the European Union (grant number: RIA2018D-2498; TB TRIAGE +). The funders had no role in study design, data collection and analysis, decision to publish, or preparation of the manuscript.

associated with survival, which is likely explained by the low specificity of VISITECT. Challenges with CD4 testing and preventive treatment uptake require addressing.

## Introduction

While AIDS-related deaths have plateaued worldwide, one in three people living with HIV (PLHIV) still present to care or re-engage in care with advanced HIV disease (AHD) [1]. AHD is defined in adults as having a CD4-count below 200 cells/µl, or a World Health Organization (WHO) stage 3 or 4 condition [2, 3]. An estimated 10% of PLHIV with AHD die within three months after anti-retroviral treatment (ART) initiation [2]. Up to 65% of AIDS-related deaths are attributable to tuberculosis (TB) or cryptococcal meningitis [4]. In 2017, WHO recommended a package of care to reduce AHD-related mortality, including interventions to prevent, diagnose and treat TB, cryptococcosis and severe bacterial infections, rapid ART initiation and enhanced adherence support [3, 5, 6]. Necessary diagnostic components include CD4-count testing (ideally at point-of-care) and lateral flow assays for cryptococcal antigen (CrAg) and TB lipoarabinomannan (TB-LAM) screening [3].

Sub-Saharan Africa accounted for 65% of global AIDS-related deaths in 2021 [7]. Yet, the AHD care package is poorly implemented in the region [8]. The use of CD4-testing at baseline has reduced in many health care facilities following the implementation of universal test and treat guidelines allowing ART start regardless of CD4-count [9]. A point-of-care CD4-test (VISITECT CD4 Advanced Disease; VISITECT, Omega (since 2022; AccuBio), UK), produces a semiquantitative result of above or below 200 CD4 cells/µl. Two studies reported a sensitivity and specificity respectively ranging between 82–95% and 82–86% on venous blood and 98% and 77% on capillary blood for VISITECT [10, 11]. In one study, the test was found acceptable and adequately performed by different professional cadres [10].

The feasibility of implementing this package, with VISITECT as CD4-test, has not yet been evaluated. We offered AHD care package point-of-care diagnostics, including VISITECT, to PLHIV recruited during near-facility TB case-finding, referred them with test results to the health care setting, and collected data on 12-week survival and treatment status.

## Method

### Study aim

We aimed to evaluate the feasibility of AHD care package implementation considering four aspects: acceptability, intervention delivery, process compliance of point-of-care diagnostics, and early effectiveness (survival and treatment status) of the package after referral with test results. To assess process compliance and show those eligible for treatment initiation (antiretroviral treatment (ART), TB treatment, cotrimoxazole or TB preventive treatment) after referral, we first presented test results indicating AHD (CD4≤200cells/µl on VISITECT or a WHO stage 3 or 4 condition). Finally, we assessed associations between AHD and TB status and survival (being confirmed alive at 12 weeks, versus death or unknown status).

### Design, setting and population

Protocols of this prospective mixed method feasibility study and of the trial in which it was embedded are available (TB TRIAGE+ ACCURACY: ClinicalTrials.gov: NCT04666311) [12].

In 2021, the adult HIV prevalence in Lesotho and South Africa, respectively, was estimated at 21% and 18%, and 4500 and 51000 people died of AIDS [13]. In 2019, 24% percent of patients with a new HIV diagnosis in Lesotho and 31% in South Africa had a CD4-count <200cells/μl [14]. This study, and all trial procedures, were conducted in a study-specific container-based temporary clinic placed at Butha-Buthe district hospital in Lesotho and in a mobile clinic and X-ray unit based at a primary health care clinic or Human Sciences Research Council research site in the Vulindlela community, Pietermaritzburg, in KwaZulu-Natal, South Africa. In Lesotho all point-of-care tests (including AHD-related tests) were performed in a mini-lab, and in South Africa in the mobile clinic. The trial enrolled consenting adults (≥18 years) with TB symptoms (cough, fever, weight loss or night sweats of any duration) presenting to facilities in Lesotho and South Africa, between February 2021 and March 2022. Exclusion criteria were self-reported pregnancy, being critically sick and in need of immediate medical care, being on TB-treatment, or any condition for which participation in the study, as judged by the investigator, could compromise the well-being of the subject, or prevent, limit, or confound protocol specified assessments. The present study used quantitative data collected from PLHIV enrolled in the "TB TRIAGE+ ACCURACY" trial plus quantitative and qualitative data collected from i) all study staff implementing point-of-care diagnostics during this study ("implementers") ii) staff who received training on point-of-care diagnostic procedures ("trained staff").

## Intervention

After enrolment, anthropometric, demographic, and clinical data were collected. Participants with unknown HIV status were offered HIV testing and counselling per national guidelines [15, 16]. Known and newly diagnosed PLHIV were offered VISITECT CD4 Advanced Disease (Omega until 2022, then Accubio Limited, US; VISITECT) on venous blood, and urine Alere Determine TB-lipoarabinomannan (Abbott, US; AlereLAM). CrAg lateral flow assay (Immy, US; Immy CrAg) on plasma was performed in case of CD4≤200cells/μl (Fig 1).

Enrolled participants were asked to provide spot sputum samples for Xpert MTB/RIF and Xpert MTB/RIF Ultra (Cepheid, US; Xpert Ultra) and one for mycobacterial culture (MGIT). Tests for the main trial included CAD4TB (Delft Imaging System, NL), Afinion CRP assay (Alere Afinion, USA), CADCOVID (Delft Imaging System, NL) with white blood count, and a COVID-19 rapid diagnostic test (Clinicaltrials.gov nr NCT04666311). In both sites, radiologists performed CAD4TB and CAD4COVID. All other trial-related tests were performed by nurses in South Africa while in Lesotho, lay counsellors provided HIV testing and were trained to provide other tests in case of absence of nurses. On average, seven trial participants were enrolled each day per site, of whom half were PLHIV. In both sites, two nurses were working in parallel, following individual patients through all procedures. Participants were not initiated on treatment by study staff but referred into the health system with a referral letter and test results for further management following national guidelines. At 12 weeks, survival, ART and TB treatment status were assessed by phone call and facility file review. Compliance with required follow-up investigations and/or treatment informed by trial test results was assessed by facility file review.

## Framework for feasibility evaluation

Following phase three of the Medical Research Council framework for evaluation of complex interventions, this feasibility evaluation included assessment of acceptability, intervention delivery and process compliance of the point-of-care diagnostics, and early effectiveness

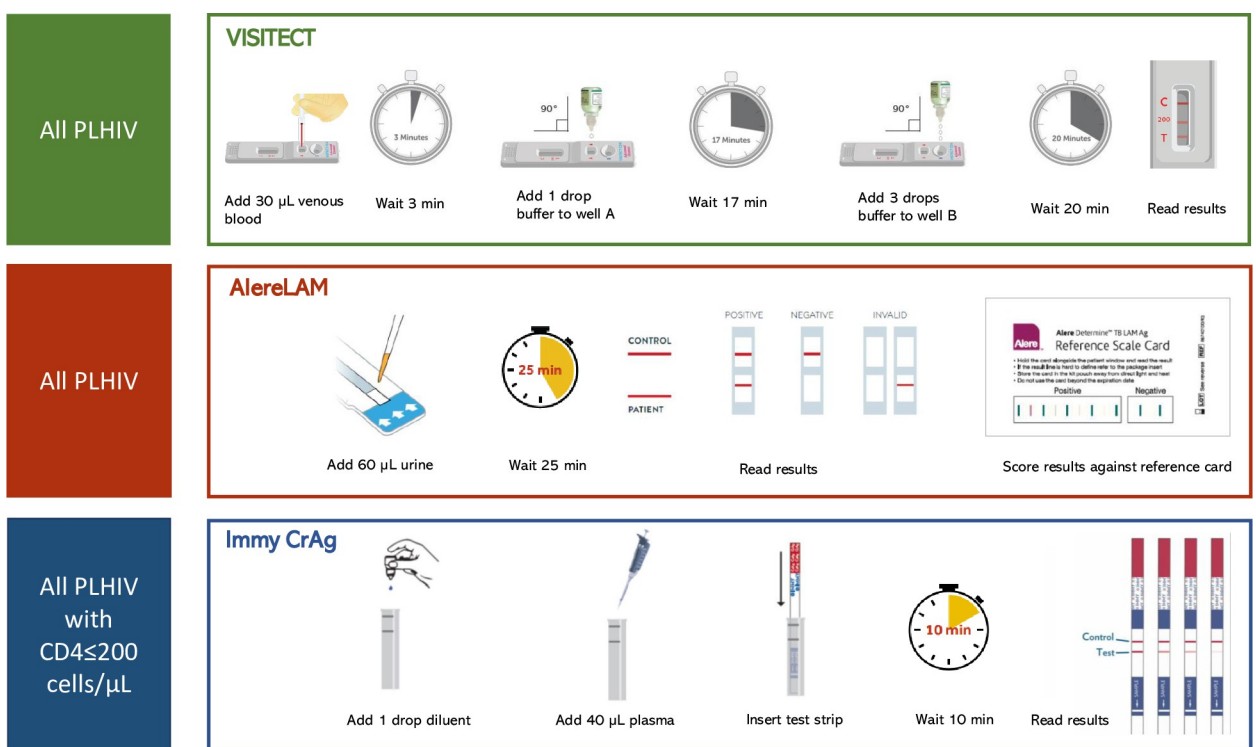

**Fig 1. Advanced HIV disease care package point-of-care test procedures in PLHIV in Lesotho and South Africa.** Immy CrAg, Immy cryptococcal antigen lateral flow assay; VISITECT; Omega VISITECT CD4 Advanced HIV; PLHIV, people living with HIV; AlereLAM, Alere tuberculosis lipoarabinomannan lateral flow assay.

(survival and treatment status) of the AHD care package after referral with test results (Fig 2 and Table 1) [17, 18].

We evaluated feasibility, by country, and by type of point-of-care test. We explored how study staff perceived the acceptability of provided services by patients and other health-care worker, inquired about challenges, enablers, and critical steps for intervention delivery during near-facility TB-case finding, and how they anticipated (future) community-based delivery of the same services.

## Variables and outcome definitions

AHD was defined as having a CD4≤200cells/µl or a WHO stage 3 or 4 condition, including any positive TB test. We constructed a composite TB test, considered positive if either Xpert, Xpert Ultra, MGIT or AlereLAM were positive, negative if all tests were negative, and unknown if not all tests were performed and none of the performed were positive. Twelve [10–14] week outcomes were defined as follows: dead if reported as dead by call or file review, alive if reported as alive by call or file review (and not dead), and unknown if no result was retrieved by call nor file review. Unfavourable outcomes were death or an unknown status. PLHIV were considered eligible for starting i) TB treatment if the composite TB test was positive, ii) ART if newly diagnosed with HIV, or known HIV-positive but not on ART, iii) cotrimoxazole if CD4≤200 cells/µl, and iv) tuberculosis preventive therapy if newly diagnosed with HIV and no positive TB test.

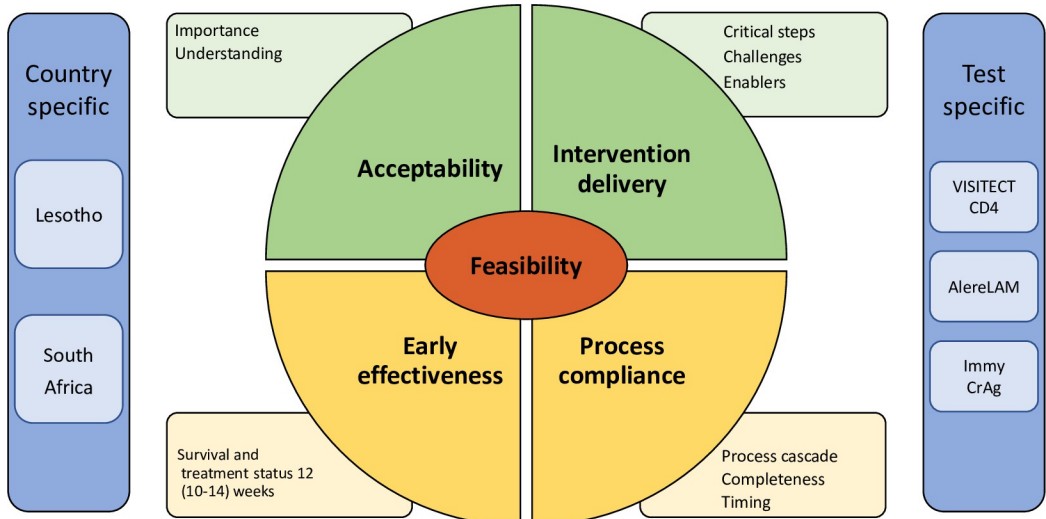

**Fig 2. Framework for evaluation of the feasibility of implementing the advanced HIV disease care package.** Pillars of the evaluation assessed with quantitative methods are shown in yellow, with mixed methods in green. Specific topics of interest for feasibility evaluation are shown in blue. AlereLAM, Alere tuberculosis lipoarabinomannan lateral flow assay. Immy CrAg, cryptococcal antigen lateral flow assay; VISITECT CD4, Omega VISITECT CD4 Advanced Disease.

## Quantitative data collection and analysis

Patient-level data were collected in MACRO trial software (MACRO version 4.11.0.0, Elsevier). Statements on acceptability and intervention delivery were rated on semi-structured questionnaires in Microsoft Excel by implementers. Emerging themes were used to inform group discussion guides (S1 and S2 Tables). Each step of a test (e.g. adding buffer), and the full procedure was observed by an expert evaluator, rated with a paper-based five-point Likert-style scales (0 = incomplete to 5 = complete) and timed (S3 Table).

**Table 1. Overview of objectives, methods, and outcomes.**

| Objective | Design | Data collection tool | Population | Outcomes |
|---|---|---|---|---|
| Describe test results | Quantitative | MACRO trial software | PLHIV | Frequencies, proportions of test results |
| Evaluate the feasibility of implementation | | | | |
| Acceptability | Quantitative | semi-structured questionnaires | Implementers | Mean (IQR) rating of agreement with statements |
| | Qualitative | group discussions | Trained staff | Understanding of the procedure, perceptions on its importance |
| Intervention delivery | Quantitative | semi-structured questionnaires | Implementers | Mean (IQR) rating of agreement with statements |
| | Qualitative | group discussions | Trained staff | Barriers, enables and critical steps |
| Process compliance | Quantitative | MACRO trial software | PLHIV | Frequencies, proportions of eligible patients receiving tests |
| | Quantitative | Likert-style rating scale | PLHIV | Rating (0–5) of completeness and duration of test procedures |
| Early effectiveness | Quantitative | MACRO trial software | PLHIV | Frequencies, proportions of those alive at 12 [10–14] weeks |
| | | | | Frequencies, proportions of eligible patients with treatment initiated at 12 [10–14] weeks |
| Explore associations of AHD and TB with outcomes | Quantitative | MACRO trial software | PLHIV | Adjusted odds ratio of being alive at 12 [10–14] weeks |

IQR, interquartile range; PLHIV, people living with HIV

Categorical variables were presented with frequencies and proportions. Independent samples were compared with the chi-squared or Fisher's exact test, as appropriate. Continuous variables were presented with medians with interquartile ranges (IQR) and compared using the Wilcoxon rank sum test. We conducted a logistic regression, selecting variables associated (p<0.2) in univariable regression to be included in multivariable models, with AHD status (model 1) and TB status (model 2) as predictors of interest, and age and sex included a priori. A p-value of <0.05 was considered statistically significant. Stata version 16.1 was used for analysis.

## Qualitative data collection and analysis

Group discussions among study staff trained on AHD procedures were held on Zoom (investigator online, participants on site, due to COVID-19 restrictions), aided with topic guides. Participants included nurses, lay counsellors, radiographers and study coordinators with varying backgrounds and type and level of experience. We used an iterative approach, adapting topic guides with emerging themes. Participants could express themselves in English, Sesotho or Isi-Zulu. Discussions were audio-recorded, transcribed, and initially coded using NVivo 1.7 (QSR International, Doncaster, Vic., Australia). After coding, data were condensed and categorized into broader themes through thematic content analysis.

## Triangulation

We applied convergent triangulation to validate information obtained from different research methods and to identify convergent results, and holistic triangulation to identify unique perspectives per method. After separating quantitative and qualitative analysis, results were mapped using the framework for feasibility evaluation (Fig 2) and analysed for convergent and unique information. Diverging components are mentioned when specific challenges were noted.

## Ethics approval and consent to participate

Protocols from this study and TB TRIAGE+ ACCURACY were approved by the National Health Research and Ethics Committee (ID 100–2020), Lesotho, the Human Sciences Research Council Research Ethics Committee (No REC 2/23/09/20 & No REC 2/23/09/20a) and Provincial Department of Health (DOH-27-022021-5641), South Africa, and the Ethikkomission Nordwest-und Zentralschweiz (AO_2020_00014 & AO_2020_00015), Switzerland. All study participants (patients and implementers) provided written informed consent.

## Results

### Patient characteristics and test results

Among 1392 participants enrolled, 48.6% (676) were PLHIV; 47.9% (335/700) among those enrolled in Lesotho and 49.3% (341/692) among South African participants. Compared to South African patients, patients from Lesotho were older, had lower BMI, and presented more often with cough, fever, or weight loss. In Lesotho and South Africa, respectively, 45.4% and 14.7% of participants (p<0.001) had VISITECT indicating CD4≤200 cells/μl, and 23.9% versus 17.3% (p<0.001) had a positive composite TB test (Table 2).

**Implementer characteristics.** All eight implementers completed two questionnaires at three and six months after study start (Table 3, results in S1 and S2 Tables). Group discussions were held a week later: two site-specific discussions with seven participants in Lesotho (GD1,

**Table 2. Characteristics and test results in PLHIV enrolled in Lesotho and South Africa, February 2021-March 2022.**

| Characteristic | | Overall (n = 676) | | Lesotho (n = 335) | | South Africa (n = 341) | | |
|---|---|---|---|---|---|---|---|---|
| | | n | % | n | % | n | % | p-value# |
| Median age (years, IQR) | | 46 | (36–54) | 46 | (36–57) | 45 | (36–52) | 0.063 |
| Age category (years) | 18–34 | 144 | 21.3 | 69 | 20.6 | 75 | 22.0 | 0.009 |
| | 35–64 | 469 | 69.4 | 229 | 68.4 | 250 | 73.3 | |
| | ≥65 | 53 | 7.8 | 37 | 11.0 | 16 | 4.7 | |
| Sex | Female | 350 | 51.8 | 170 | 50.7 | 156 | 45.7 | 0.193 |
| | Male | 326 | 48.2 | 165 | 49.3 | 185 | 54.3 | |
| HIV status | Newly diagnosed | 53 | 7.8 | 35 | 10.4 | 18 | 5.3 | <0.001 |
| | Known positive on ART | 583 | 86.2 | 265 | 79.1 | 318 | 93.3 | |
| | Known positive not on ART | 40 | 5.9 | 35 | 10.4 | 5 | 1.5 | |
| Median BMI (kg/m2, IQR) | | 22 | (19–27) | 22 | (18–25) | 22 | (20–28) | 0.004 |
| BMI category (kg/m2) | <18.5 | 149 | 22.0 | 89 | 26.6 | 60 | 17.6 | 0.009 |
| | 18.5–24 | 319 | 47.2 | 156 | 46.6 | 163 | 47.8 | |
| | ≥25 | 208 | 30.8 | 90 | 26.9 | 118 | 34.6 | |
| TB symptoms | Fever | 151 | 22.3 | 101 | 30.1 | 50 | 14.7 | <0.001 |
| | Cough | 616 | 91.1 | 318 | 94.9 | 298 | 87.4 | 0.001 |
| | Night sweats | 309 | 45.7 | 142 | 42.4 | 167 | 49.0 | 0.086 |
| | Weight loss | 368 | 54.4 | 198 | 59.1 | 170 | 49.9 | 0.016 |
| **Test results** | | | | | | | | |
| VISITECT CD4 Advanced Disease | ≤200cells/µl | 202 | 29.9 | 152 | 45.4 | 50 | 14.7 | <0.001§ |
| | >200cells/µl | 470 | 69.5 | 180 | 53.7 | 290 | 85.0 | |
| | Not done | 4 | 0.6 | 3 | 0.9 | 1 | 0.3 | |
| AlereLAM | Positive | 89 | 13.2 | 54 | 16.1 | 35 | 10.3 | 0.021§ |
| | Negative | 585 | 86.5 | 281 | 83.9 | 304 | 89.1 | |
| | Not done | 2 | 0.3 | 0 | 0.0 | 2 | 0.6 | |
| Xpert MTB/RIF | MTB detected | 62 | 9.2 | 34 | 10.1 | 28 | 8.2 | <0.001§ |
| | MTB not detected | 591 | 87.4 | 299 | 89.3 | 292 | 85.6 | |
| | Unknown, undetermined, or not done | 23 | 3.4 | 2 | 0.6 | 21 | 6.2 | |
| Xpert MTB/RIF Ultra | MTB detected | 67 | 9.9 | 37 | 11.0 | 30 | 8.8 | <0.001§ |
| | MTB not detected | 592 | 87.6 | 297 | 88.7 | 295 | 86.5 | |
| | Unknown, undetermined, or not done | 17 | 2.5 | 1 | 0.3 | 16 | 4.7 | |
| MGIT culture | Positive for MTB | 59 | 8.7 | 39 | 11.6 | 20 | 5.9 | <0.001§ |
| | Negative for MTB | 538 | 79.6 | 284 | 84.8 | 254 | 74.5 | |
| | Contaminated | 9 | 1.3 | 9 | 2.7 | 0 | 0.0 | |
| | Unknown, not done, or NTM | 70 | 10.4 | 3 | 0.9 | 67 | 19.6 | |
| Composite TB test | Positive† | 139 | 20.6 | 80 | 23.9 | 59 | 17.3 | <0.001 |
| | Negative | 464 | 68.6 | 244 | 72.8 | 220 | 64.5 | |
| | Unknown | 73 | 10.8 | 11 | 3.3 | 62 | 18.2 | |
| Advanced HIV disease status | AHD* | 282 | 41.7 | 183 | 54.6 | 99 | 29.0 | <0.001 |
| | No AHD | 333 | 49.3 | 144 | 43.0 | 189 | 55.4 | |
| | Unknown | 61 | 9.0 | 8 | 2.4 | 53 | 15.5 | |
| Immy CrAg LFA (among ≤200cells/µl) | Positive | 8 | 4.0 | 4 | 2.6 | 4 | 8.0 | 0.109§ |
| | Negative | 192 | 95.0 | 146 | 96.1 | 46 | 92.0 | |

(*Continued*)

**Table 2.** (Continued)

| | | Overall (n = 676) | | Lesotho (n = 335) | | South Africa (n = 341) | | |
|---|---|---|---|---|---|---|---|---|
| | Not done | 2 | 1.0 | 2 | 1.3 | 0 | 0.0 | |

\# chi-squared or Fisher's exact test for categorical variables, wilcoxon signed rank test for continuous variables

†MTB detected on Xpert MTB/RIF, Xpert MTB/RIF Ultra or MGIT culture or AlereLAM

*CD4 ≤200cells/μl or AlereLAM positive or MTB detected on Xpert MTB/RIF or Xpert MTB/RIF Ultra or MGIT culture

AHD, advanced HIV disease; AlereLAM, Alere tuberculosis lipoarabinomannan lateral flow assay; CI, confidence interval; Immy CrAg, Immy cryptococcal antigen lateral flow assay; MGIT, Mycobacteria growth indicator tube; VISITECT, VISITECT CD4 Advanced Disease

93 minutes) and four in South Africa (GD2, 125 minutes) and one joint discussion with 10 trained staff, including all implementers (GD3, 130 minutes).

## Acceptability

**For study staff.** Acceptability of implementation was high among study staff, and perceived high for other healthcare workers and patients. This was partially since AHD was seen as an important problem in both countries among newly diagnosed and ART-experienced PLHIV. Reasons for reported AHD were geographical and financial barriers to access services, and ART-related stigma.

*"People already on ART sometimes interrupt treatment due to lack of money to reach services. When they return to care, they have AHD."* (GD1, Lesotho)

The AHD care package was considered an important intervention to prevent mortality, for instance from cryptococcal meningitis, and an important tool for healthcare workers.

*"*[The package helps] *to know if they have cryptococcal meningitis, that is one of the leading causes of death in PLHIV. So that we know if the patient needs fluconazole, for their system to pick up and survive."* (GD3, Lesotho)

*"AHD* [care package] *is really helping out providers to identify patients that were likely to be lost, to lose their lives."* (GD3, Lesotho).

**Table 3. Characteristics of advanced HIV disease care package implementers.**

| Characteristics | | All | Lesotho | South Africa |
|---|---|---|---|---|
| Profile | Nurse | 5 | 2 | 3 |
| | Lay counsellor | 3 | 3 | 0 |
| Median age (IQR) | | 35 (30–41) | 32 (27–35) | 36 (24–56) |
| Sex | Female | 6 | 4 | 2 |
| | Male | 2 | 1 | 1 |
| Education | Secondary | 3 | 3 | 0 |
| | Tertiary | 5 | 2 | 3 |
| Work experience (median years, IQR) | | 5 (4–12) | 5 (4–5) | 14 (4–20) |
| AlereLAM experience (n, years) | | 1, 1 | 0 | 1 |
| Immy CrAg experience (n, median years, IQR) | | 4, 5 (4–5) | 4, 5 (4–5) | 0 |

AlereLAM, Alere tuberculosis lipoarabinomannan lateral flow assay, Immy CrAg, Immy cryptococcal antigen lateral flow assay; IQR, interquartile range

There was a perception that without the package, AHD would not be communicated with patients, as in usual care CD4-results are not explained. As CD4 is not done at point-of-care, patients are often no longer present when results are available.

*"[I would recommend the package] because they will know their CD4 through VISITECT on the spot. At the clinic they won't explain what happens to them."* (GD1, South Africa)

**For other healthcare workers.** Staff were concerned that public sector healthcare workers might perceive the package as additional workload, which could be countered by sufficient training, and by explaining them: *"It's going to be a lot of work* [on] *your side, but it's going to help you to give quality care rather than quantity."* (GD3, Lesotho)

**For patients.** Trained staff perceived that acceptability was high for patients. A barrier to accepting the package was reported when a patient refused further testing after being diagnosed with HIV. *"Getting this news for the first time, she couldn't stand anything, she wanted to be on her own."* (GD1, South Africa)

**For community-based implementation.** There was agreement that the package should be implemented widely, including in community-based outreach activities, health posts, testing campaigns, and house-to-house HIV or TB-case finding.

*"I think the package can be given anywhere and would benefit all people who it is administered to. Even house-to-house is possible, because we could otherwise be excluding those who cannot attend gatherings, like old people, those living far from the gathering or newly feeding mothers. If we go to grassroots level, it could really help all those that need it."* (GD1, Lesotho)

*"It* [the package] *should be implemented for primary healthcare, in clinics, communities and outreaches because it is the basis of* [responding to] *all patient's needs."* (GD3, South Africa)

## Intervention delivery

Implementers reported being comfortable with the full procedure after delivering it to a median of 2 (IQR:1–3) patients.

**Challenges.** Among challenges to implementation overall, the most prominent ones reported were stockouts and short expiry dates of VISITECT in Lesotho, the overall long procedure, and interpretation of all different tests results. Initially, the South African team, contrary to the Lesotho team, did not run the VISITECT and AlereLAM tests in parallel, leading to a longer procedure.

*"Stockouts is the biggest problem we encountered since we started AHD* [care package]. *Also expiry dates, short shelf lives."* (GD1, Lesotho)

*"When you do VISITECT, you need to do TB-LAM as well. If you do that simultaneously, you can save time."* (GD2, South Africa)

In South Africa, clinics were closed due to riots for ten days in July 2021, which led to an interruption of the trial. The team saw availability of point-of-care tests as an opportunity to ensure continuity of testing: *"When the actual machines that do the analysis are looted, broken, and burned down and you have little portable point-of-care tests and the results are accurate, it's*

*another way to at least get going until the gold standard of testing is sorted and fixed."* (GD2, South Africa)

Both teams said they worked more efficiently and were more confident about results interpretation with time. During GD3, when stockouts had been solved and riots had finished, when asked if there were still difficulties with the procedure, the answer was: *"We are cruising!"* (GD3, Lesotho)

**Enablers.** The teams emphasized the need for practical face-to-face training, and job aids, which could enable different cadres, including counsellors, to conduct the procedures. In both countries, the legal scope of practice of counsellors prohibits them to perform phlebotomy. The staff thus suggested to use finger prick sampling for VISITECT and Immy CrAg in case counsellors perform the tests.

In South Africa, the team initially used a wrong sample for CrAg (whole blood instead of plasma) because it was not clear from the online training. *"For CrAg training, the trainer was not visible, so it was not clear which materials were being used. The trainer needs to be on site to do the practical."* (GD3, South Africa)

*"Almost everyone can do* [the AHD care package] *with training and job aids and remembering the timing*: *lab persons*, *nurses*, *lay counsellors*, *doctors."* (GD3, Lesotho).

Counselling, including letting patients interpret test results, was deemed essential to enable patients to understand the need for the long procedures, and the need for further testing in case of a new HIV diagnosis.

**Critical steps.** Specific difficulties with VISITECT were the many procedural steps, the overall long duration, results interpretation, and the fact that it does not produce an exact CD4-result but shows whether it is below or above 200cells/µl.

*"Sometimes you wonder whether your eyes are being honest with you, when the light changes you compare with the reference, so you need a second opinion."* (GD1, Lesotho)

*"*[VISITECTs] *main disadvantage is its duration, and it's not precise for the CD4-count."* (GD2, South Africa)

For AlereLAM, critical steps mentioned were sampling, as implementers could not verify the production of midstream urine by participants, and results reading and rating. Uncertainties about result interpretation led to fears about wrong TB treatment initiation in future implementation in the community.

*"We will not be in the loo* [i.e. toilet] *when it is the time* [to provide the midstream urine sample]. *And some will tell you they don't have breaks. It will start and stop and then. . ."* (GD2, South Africa)

*"You could find that TB-LAM was negative while it was read to be positive and initiate a patient on TB treatment who should not have been initiated."* (GD1, Lesotho)

A lack of buffer and diluent led to people not being tested with Immy CrAg in Lesotho, and implementers suggested the manufacturer should provide extra buffer.

## Process compliance

Compliance with each AHD point-of-care diagnostic test was ≥99% (Fig 3).

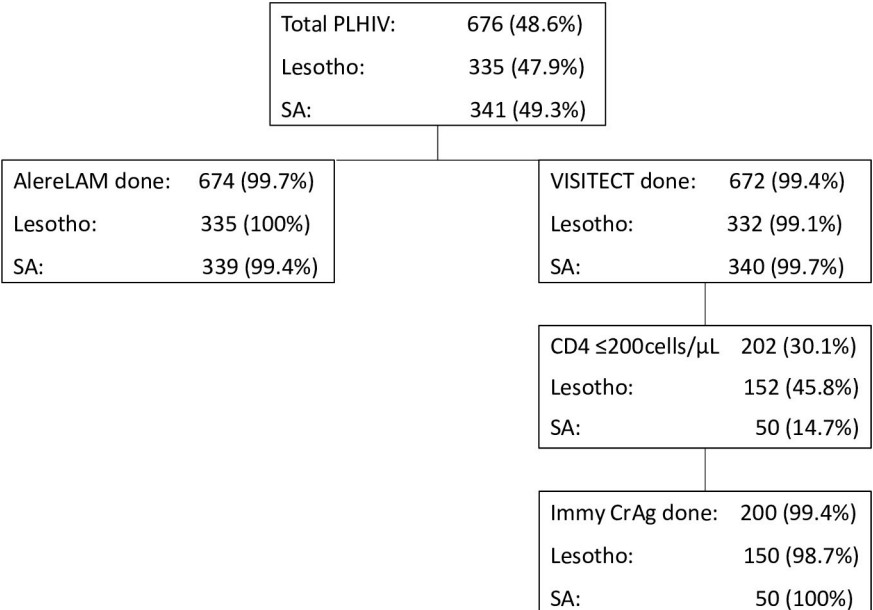

**Fig 3. Process cascade for advanced HIV disease care package in PLHIV with TB symptoms.** Total PLHIV: known positive and newly tested. AlereLAM, Alere tuberculosis lipoarabinomannan lateral flow assay; Immy CrAg, Immy cryptococcal antigen lateral flow assay; SA, South Africa VISITECT, VISITECT CD4 Advanced Disease.

Reasons for not performing tests were refusal after HIV diagnosis (1 VISITECT and Alere-LAM), failure to produce a sample (1 AlereLAM), and stockouts of tests (3 VISITECT) or reagents (2 Immy CrAg). The median minutes to perform VISITECT (n = 12) was 45 (IQR: 42 −51), AlereLAM (n = 12) 34 (IQR: 31−37), Immy CrAg (n = 4) 11 (IQR:10−13) and all three tests (n = 4) 73 min (IQR: 68−85) (Rating scale results: S3 Table).

## Early effectiveness

Eleven patients (1.6% of 676) were not contacted within the 12-week window and excluded. Outcomes at 12 weeks among 665 PLHIV who were contacted are presented in Table 4.

Among the 50 newly diagnosed HIV-positive patients, among the 578 known to be HIV positive and on ART, and among the 37 known HIV positive patients who had interrupted ART, respectively 2.0% (n = 1), 1.4% (n = 8) and 8.1% (n = 3) had died at 12 weeks. Treatment status at 12 weeks among those eligible is presented in Table 5.

In multivariable logistic regression, having AHD (versus not, model 1: aOR: 0.71 (95%CI: 0.40−1.27), or having a positive composite TB test (versus negative, model 2: aOR: 1.10 (95% CI: 0.59−2.07)) were not associated with being alive at 12 weeks. Participants from Lesotho (vs South Africa) had lower odds (model 1: aOR: 0.45 (95% CI 0.25−0.80), model 2: aOR: 0.44 (95% CI 0.25−0.77) of being alive at 12 weeks in both models (S4 Table). Among seven alive CrAg-positive PLHIV, three with a positive TB test had been initiated on TB treatment.

## Discussion

This is the first feasibility study on the implementation of an AHD care package with VISI-TECT to identify a CD4≤200 cells/μl. Implementation of the point-of-care diagnostic tests was feasible during a TB case-finding study and led to acceptable effectiveness in two high HIV- and AHD-burden settings: 90% of all PLHIV, and 85% of those with AHD, were alive

**Table 4.  Twelve-week survival of PLHIV after enrolment in Lesotho and South Africa, February 2021-March 2022.**

| | | | Dead | | Unknown | | Vital status | | | | |
| | | | | | | | Unfavourable outcome * | | Alive | | |
| Country / Test results | | Total | n | % | n | % | n | % | n | % | p-value # |
| All participants | | 665 | 12 | 1.8 | 58 | 8.7 | 70 | 10.5 | 595 | 89.5 | |
| Country | Lesotho | 324 | 10 | 3.1 | 39 | 12.0 | 49 | 15.1 | 275 | 84.9 | <0.001 $ |
| | South Africa | 341 | 2 | 0.6 | 19 | 5.6 | 21 | 6.2 | 320 | 93.8 | |
| AHD status | AHD† | 276 | 9 | 3.3 | 33 | 12.0 | 42 | 15.2 | 234 | 84.8 | 0.002 |
| | No AHD | 328 | 3 | 0.9 | 23 | 7.0 | 26 | 7.9 | 302 | 92.1 | |
| | Unknown | 61 | 0 | 0.0 | 2 | 3.3 | 2 | 3.3 | 59 | 96.7 | |
| VISITECT | CD4≤200cells/μl | 196 | 8 | 4.1 | 25 | 12.8 | 33 | 16.8 | 163 | 83.2 | 0.001 |
| | CD4>200cells/μl | 465 | 4 | 0.9 | 32 | 6.9 | 36 | 7.7 | 429 | 92.3 | |
| | Unknown | 4 | 0 | 0.0 | 1 | 25.0 | 1 | 25.0 | 3 | 75.0 | |
| Composite TB test | Positive‡ | 138 | 2 | 1.4 | 15 | 10.9 | 17 | 12.3 | 121 | 87.7 | 0.047 |
| | Negative | 454 | 10 | 2.2 | 41 | 9.0 | 51 | 11.2 | 403 | 88.8 | |
| | Unknown | 73 | 0 | 0.0 | 2 | 2.7 | 2 | 2.7 | 71 | 97.3 | |
| Immy CrAg (among ≤200cells/μL) | Positive | 8 | 1 | 12.5 | 0 | 0.0 | 1 | 12.5 | 7 | 87.5 | 0.838 |
| | Negative | 187 | 7 | 3.7 | 26 | 13.9 | 33 | 17.6 | 154 | 82.4 | |
| | Unknown | 1 | 0 | 0.0 | 0 | 0.0 | 0 | 0.0 | 1 | 100.0 | |

Comparison of unfavourable outcomes with being alive (with and without unknown test results), with

# Fisher's exact test, or

$ chi-squared test.

* Unfavourable outcomes are dead or unknown status

† CD4 ≤200cells/μL or AlereLAM positive or MTB detected on Xpert MTB/RIF or Xpert MTB/RIF Ultra or MGIT culture

‡ MTB detected on Xpert MTB/RIF, Xpert MTB/RIF Ultra or MGIT culture or AlereLAM

AHD, advanced HIV disease; AlereLAM, Alere tuberculosis lipoarabinomannan lateral flow assay; CI, confidence interval; Immy CrAg, Immy cryptococcal antigen lateral flow assay; MGIT, Mycobacteria growth indicator tube; PLHIV, people living with HIV; VISITECT, VISITECT CD4 Advanced Disease

three months after receiving the AHD care package diagnostics. False positive VISITECT results likely contributed to the high apparent AHD prevalence and survival.

Having a CD4≤200 cells/μl and AHD were highly prevalent in our study, especially in Lesotho. In previous populations surveys, 11% among recently diagnosed PLHIV Butha-Buthe, and 9% among PLHIV in KwaZulu Natal had a CD4≤200 cells/μl [19, 20]. The hospital setting in Lesotho attracted sicker patients as compared to the primary health clinic near the mobile unit in South Africa, and the former were thus more likely to have AHD. We used VISITECT operated by nurses, outside of a clinical laboratory. In two published studies, VISITECT was reported to have good diagnostic accuracy on venous blood, with laboratory technicians as operators: sensitivity was 95% (95% CI 91–98) and 94% (95% CI 88%–98%), and specificity 82% (95% CI 78–85%) and 86% (95% CI 84%–88%) [10, 11]. On capillary blood, with different cadres conducting the test, 98% (95% CI 95–100.0) sensitivity, and 77% (95% CI 72–82%) specificity was reported [10]. However, in both studies, false positive misclassification (i.e. as CD4≤200 cells/μl) of patients with a CD4 between 200–350 cells/μl was common [10, 11, 21]. This could help explain the high prevalence of CD4≤200 cells/μl in our study, and the relatively low Immy CrAg positivity in Lesotho. A meta-analysis pooled prevalence of CrAg positivity in PLHIV with CD4<200 cells/μL was 5% (95% CI 2–7), and 6.5% was previously reported in the study district in KwaZulu Natal [22, 23]. AlereLAM-positivity in our study was in line with 17% reported in a five-country study in a population with a similar characteristics

**Table 5. Twelve-week treatment status among PLHIV eligible for initiation of different (preventive) treatments.**

| | Total | Dead | | Unknown | | Total alive | | Alive | | | | | | |
| | | | | | | | | On correct treatment | | Not on treatment | | Unknown treatment status | |
| | | n | % | N | % | n | % | n | % | n | % | n | % |
| **All participants** | 665 | 12 | 1.8 | 58 | 8.7 | 595 | 89.5 | NA | | NA | | NA | |
| **Eligible for antiretroviral treatment** | 87 | 4 | 4.6 | 9 | 10.3 | 74 | 85.1 | 56 | 75.7 | 8 | 10.8 | 10 | 13.5 |
| *Among whom*: | | | | | | | | | | | | | |
| Newly HIV positive | 50 | 1 | 2.0 | 4 | 8.0 | 45 | 90.0 | 35 | 77.8 | 8 | 17.8 | 2 | 4.4 |
| Known HIV interrupted ART | 37 | 3 | 8.1 | 5 | 13.5 | 29 | 78.4 | 21 | 72.4 | 0 | 0.0 | 8 | 27.6 |
| *Among whom*: | | | | | | | | | | | | | |
| AHD* | 68 | 3 | 4.4 | 8 | 11.8 | 57 | 83.8 | 49 | 86.0 | 2 | 3.5 | 6 | 10.5 |
| No AHD | 15 | 1 | 6.7 | 1 | 6.7 | 13 | 86.7 | 6 | 46.2 | 3 | 23.1 | 4 | 30.8 |
| Unknown | 4 | 0 | 0.0 | 0 | 0.0 | 4 | 100.0 | 1 | 25.0 | 3 | 75.0 | 0 | 0.0 |
| **Eligible for TB treatment** | 138 | 2 | 1.4 | 15 | 10.9 | 121 | 87.7 | 95 | 78.5 | 5 | 4.1 | 21 | 17.4 |
| **Eligible for cotrimoxazole** | 196 | 8 | 4.1 | 25 | 12.8 | 163 | 83.2 | 48 | 29.4 | 55 | 33.7 | 60 | 36.8 |
| **Eligible for TB preventive therapy** | 28 | 1 | 3.6 | 2 | 7.1 | 25 | 89.3 | 5 | 20.0 | 8 | 32.0 | 12 | 48.0 |

*CD4 ≤200cells/μL or AlereLAM positive or MTB detected on Xpert MTB/RIF or Xpert MTB/RIF Ultra or MGIT culture

† MTB detected on Xpert MTB/RIF, Xpert MTB/RIF Ultra or MGIT culture

AlereLAM, Alere tuberculosis lipoarabinomannan lateral flow assay; CI, confidence interval; Immy CrAg, Immy cryptococcal antigen lateral flow assay; MGIT, Mycobacteria growth indicator tube

[24]. In the ongoing phase of the TB TRIAGE+ study, a batch recall confirmed suboptimal specificity of one VISITECT batch. In a separate study, we are assessing the diagnostic accuracy of VISITECT in more pragmatic circumstances compared to published studies (Clinical-Trials.gov Identifier: NCT04089423).

Our study staff found the AHD care package diagnostics acceptable, despite challenges. Among them was the long procedural duration; a median of 73 minutes to conduct VISITECT, AlereLAM and Immy CrAg. While the total time was reduced by running TB-LAM and VISI-TECT in parallel to a minimum recorded time 52 minutes, it was still considered a challenge. In a setting where PLHIV would be recruited regardless of TB symptoms, a CD4 below 200 cells/μl would be conditional to AlereLAM and CrAg testing, which would prolong the duration even more (AlereLAM takes 25 minutes, while CrAg 10 minutes). Interviewed staff also thought the procedure would be acceptable for other health-care workers, despite an anticipated increased workload, linked to this long procedure. They considered that any cadre can perform these tests with job aids, after face-to-face training. The procedure was also perceived acceptable for patients by study staff, who emphasized that appropriate counseling should accompany the testing procedures. To address this counselling need and considering the increased workload and long procedure, we concur with Ndlovu et al. that a dedicated lay counsellor would ideally be appointed to provide these tests [10]. In our setting, only clinicians were allowed to draw venous blood for VISITECT and Immy CrAg, per study protocol. Both these tests can also be performed on capillary blood, so even in similar conditions, lay counsellors could perform the tests using finger prick samples [10, 23]. Process compliance for the three AHD-related point-of-care tests was excellent and reasons for missing tests confirmed the qualitative results. High compliance with VISITECT process steps was reported previously [10].

We experienced VISITECT stockouts, due to supply delays and a short shelf life of one-year. Additional VISITECT-related challenges were correct result reading, and compliance with procedural steps, echoing previous reports [10]. Errors which might have occurred

during these steps could also have led to false positive results [10, 11]. While some existing CD4-test manufacturers have stopped production, point-of-care CD4-tests with shorter and easier testing procedures and longer shelf-lives compared to VISITECT are needed [9, 25]. The fact that VISITECT does not produce an exact CD4-count was also seen as a limitation. As CrAg testing and cotrimoxazole initiation depend on different thresholds than 200 cells/µl, and clinical management might differ depending on CD4 counts, complementary roles remain necessary for semi-quantitative and quantitative CD4-tests [4, 25]. Critical steps reported for AlereLAM in our study were correct sampling and results interpretation, confirming results from three low-resource settings [26]. Bacterial contamination can cause false positive Alere-LAM, the risk of which reduces by adherence to midstream urine collection [27]. The novel urine Fujifilm SILVAMP TB-LAM is more robust against sampling error, with a higher diagnostic accuracy compared to AlereLAM, but lot-dependent variability limits its current application [24, 28]. Moreover, its duration of one hour, and five-step procedure might limit its feasibility to be integrated in the AHD care package [29]. The staff considered Immy CrAg easy to implement, but many had previous experience [30]. In Uganda, sufficient supplies and dedicated staff were enablers for CrAg testing, like in our study. In Uganda, the turn-around-time of laboratory-based CD4-results to trigger CrAg testing was a challenge [31]. In Mozambique, AHD care package implementation was also feasible during TB active case-finding, with laboratory-based CD4 and CrAg testing [32]. In our study, point-of-care testing offers additional benefits: relative ease of manipulation, transportation, fast turn-around time, and limited space requirements. As suggested by our staff, when faced with violent riots, point-of-care tests could even assist to ensure continuity of services, in case of difficult laboratory access.

In our study 90% of all PLHIV, and 85% of those with AHD, were confirmed alive three months after enrollment. Most deaths occurred among AHD patients (9/12), in Lesotho (10/12), and more frequently in those who had interrupted ART. While having AHD, having TB, and ART status were not associated with survival, being recruited in Lesotho was. Patients there presented sicker, compared to South Africa, and baseline illness was probably not completely accounted for in multivariable analysis. Lesotho patients probably also visited more from other districts, or from South Africa, making follow-up more difficult, as shown by the higher proportion of PLHIV with unknown outcome status. ART and TB-treatment initiation was confirmed in 76% and 79% of eligible patients, respectively. This is not outstanding, but the fact that AHD nor TB were associated with outcomes suggests that maybe more patients were initiated than shown in our database. We did not collect evidence on whether recommended lumbar punctures were performed after a positive CrAG test, but 7/8 CrAg-positive patients were alive at three months. ART-initiation was confirmed in 86% of 57 AHD-patients, and 46% of 13 PLHIV without AHD. This might reflect a sense of urgency to (re)start ART felt by PLHIV and healthcare providers when confronted with AHD. In Mozambique, in comparison, all eligible patients initiated ART after accompanied referral for same-day facility-based initiation [32]. Compliance with cotrimoxazole and TB preventive treatment initiation was very low in our study, which is common, and often results from barriers like stockouts and provider-hesitancy [33]. Same-day on-the-spot initiation of at least ART and preventive therapy, and improved referral after diagnosis, could help bridge this treatment gap.

Programmatic uptake of the AHD care package in Africa is very low. Among 25 African countries, only twelve recently reported that CD4-testing was widely available in ART sites, one country (South Africa) had urine TB-LAM available, and seven CrAg testing [34, 35]. We implemented the diagnostics within a pragmatic TB case-finding study, with resources not commonly available in the sub-Saharan African public sector. Our study revealed that adequate face-to-face training, quality counselling, robust supply and solid referral systems were

necessary to implement the point-of-care diagnostics successfully. Programmatic roll-out needs thus to be accompanied by sufficient funding to support implementation, including for dedicated staff. Existing differentiated care models could also serve as platforms to provide point-of-care tests [8, 34]. In places with well-established laboratory services with established supply lines and acceptable turn-around-times, point-of-care diagnostics could be preserved for remote community-based testing. As challenges with VISITECT encountered in our study could be exacerbated during public sector pragmatic implementation, evaluation of its real-life diagnostic accuracy is warranted.

Our study has important limitations. Due to the high prevalence of CD4≤200 cells/μl identified by VISITECT our study, concerns about VISITECT's diagnostic accuracy emerged. Due to the suspected low specificity of VISITECT in the current study the prevalence of CD4≤200 cells/μl and AHD is probably overestimated, and the association between AHD and survival attenuated Qualitative data were collected from study staff, recruited, and trained to perform the AHD diagnostic tests, and thus potentially more prone to give positive feedback than public sector health care workers. There might also have been social desirability bias among study staff to report positivity on tests used in the trial during group discussions. To reduce this bias, the qualitative researcher was someone not involved in their daily work, nor in evaluating their performance. Several procedures were conducted online, which can have influenced discussion dynamics. Yet, most qualitative opinions were echoed by interviewees from different cadres and settings and were confirmed with quantitative data. We also did not assess acceptability for patient directly. We implemented the AHD care package in a study with controlled conditions, which likely contributed to high testing compliance. We did not initiate patients on treatment ourselves and relied on phone calls and file reviews in the nearby clinic for follow-up, adding to missing outcomes as some patients may have sought care elsewhere.

Among the strengths of our study are the rigorous data collection for diagnostic tests, the two-country setting, and the use of multiple methods and study populations to answer our research questions. Future studies should provide longer-term prospective cohort follow-up. Patient and public health care worker perspectives on feasibility should be collected. Robust data from routine settings of the AHD care package are required to demonstrate real-world feasibility.

## Conclusions

Implementation of point-of-care diagnostics of the AHD care package, with VISITECT to identify a CD4 below 200 cells/μl, was feasible during a TB case-finding study. PLHIV had acceptable outcomes three months after referral, although the prevalence of AHD was likely overestimated due to limited VISITECT accuracy. Challenges during point-of-care testing included stockouts, long duration of procedures, and results readings, mainly related to VISITECT. However, implementers found the procedure acceptable and testing compliance was excellent. Uptake of cotrimoxazole and TB prophylactic treatment was suboptimal. For pragmatic implementation of the AHD care package with point-of-care testing to be successful, appropriately trained dedicated staff, accurate point-of-care CD4 tests, uninterrupted supplies and solid referral systems should be foreseen.

## Supporting information

**S1 Table. Agreement of implementers with statements on the advanced HIV disease care package.**
(DOCX)

**S2 Table. Difficulty of different steps of procedure perceived by implementers of advanced HIV disease care package.**
(DOCX)

**S3 Table. Summary of procedural completeness and duration of rapid tests of the advanced HIV disease care package.**
(DOCX)

**S4 Table. Logistic regression of predictors for being alive at 12 weeks.**
(DOCX)

## Acknowledgments

We thank all study participants and the staff of Solidarmed Lesotho and the Human Sciences Research Council South Africa who contributed to this study.

## Author Contributions

**Conceptualization:** Tinne Gils, Thulani Ngubane, Philip Joseph, Klaus Reither, Irene Ayakaka, Lutgarde Lynen, Alastair Van Heerden.

**Data curation:** Tinne Gils, Moniek Bresser.

**Formal analysis:** Tinne Gils, Moniek Bresser.

**Funding acquisition:** Klaus Reither, Alastair Van Heerden.

**Investigation:** Tinne Gils, Thulani Ngubane, Philip Joseph, Klaus Reither, Irene Ayakaka, Lutgarde Lynen, Alastair Van Heerden.

**Methodology:** Tinne Gils, Klaus Reither, Erika Vlieghe, Tom Decroo, Lutgarde Lynen.

**Project administration:** Mashaete Kamele, Thandanani Madonsela, Shannon Bosman, Thulani Ngubane, Philip Joseph, Irene Ayakaka, Alastair Van Heerden.

**Supervision:** Tinne Gils, Mashaete Kamele, Thandanani Madonsela, Shannon Bosman, Thulani Ngubane, Philip Joseph, Moniek Bresser, Irene Ayakaka, Lutgarde Lynen, Alastair Van Heerden.

**Validation:** Tinne Gils, Tom Decroo, Lutgarde Lynen.

**Visualization:** Tinne Gils.

**Writing – original draft:** Tinne Gils.

**Writing – review & editing:** Tinne Gils, Mashaete Kamele, Thandanani Madonsela, Shannon Bosman, Thulani Ngubane, Philip Joseph, Klaus Reither, Moniek Bresser, Erika Vlieghe, Tom Decroo, Irene Ayakaka, Lutgarde Lynen, Alastair Van Heerden.

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
