## [Decision Letter · Decision Letter 0]

23 Oct 2023

PONE-D-23-29770Implementation of the advanced HIV disease care package with point-of-care CD4 testing during tuberculosis case finding: a mixed-methods evaluation.PLOS ONE

Dear Gils,

Thank you for submitting your manuscript to PLOS ONE. After careful consideration, we feel that it has merit but does not fully meet PLOS ONE’s publication criteria as it currently stands. Therefore, we invite you to submit a revised version of the manuscript that addresses the points raised during the review process.

We look forward to receiving your revised manuscript.

Kind regards,

Lynn S Zijenah, PhD

Academic Editor

PLOS ONE

Journal Requirements:

Additional Editor Comments:

Under "Data Availability" the Authors stated "No - some restrictions will apply". Authors please elaborate the restrictions.

Reviewers' comments:

Reviewer's Responses to Questions

**Comments to the Author**

1. Is the manuscript technically sound, and do the data support the conclusions?

Reviewer #1: Yes

Reviewer #2: Yes

2. Has the statistical analysis been performed appropriately and rigorously? 

Reviewer #1: Yes

Reviewer #2: Yes

3. Have the authors made all data underlying the findings in their manuscript fully available?

Reviewer #1: Yes

Reviewer #2: Yes

4. Is the manuscript presented in an intelligible fashion and written in standard English?

Reviewer #1: Yes

Reviewer #2: Yes

5. Review Comments to the Author

Reviewer #1: This feasibility study on the implementation of Advanced HIV disease care package on 1392 PLHIV was very well conducted. The outcome of this study will be useful for the country programs when AHD is getting implemented.

Reviewer #2: In this implementation science study, Gils et al implement the AHD package of care into a TB case finding setting.

Overall this study is well conducted and well explained.

1. I would recommend the authors add further detail regarding the characteristics of the clinic and laboratories that wer e included in this study.

Study setting is only briefly mentioned in lines 93-94.

Please clarify what a "container-based clinic" refers to. And what is meant by a 'mobile unit' (line 94).

What types of clinics were these. On average how many patients are seen per day/week/year. What type of laboratory did they have. How many lab staff were available, and what other types of tests were being performed at the same time? This will help us to better understand implementation of the AHD package of care in context.

Are these busy, high volume clinics/labs that run many other tests? Or are these rural, low volume sites where they have a dedicated lab worker to run the Visitect test?

2. I strongly suspect that you have inaccurate Visitect results (meaning that there are many people with CD4>200 who were misclassified as having a CD4<200), as the authors have suggested in the Discussion. Which would explain high survival (low mortality) among the population with AHD.

As such, I would recommend softening your conclusions (in the paper and in the abstract), accounting for this critical limitation. Specifically in lines 47-48: "AHD nor TB were associated with survival suggesting adequate management referral". I think the lack of association with reduced survival suggests false positive Visitect results contaminating your AHD population. I would add that to your abstract. I find the wording of lines 47-48 to be confusing, with the word "nor" but "neither".

I think the issue of inaccurate Visitect results is so important in accounting for any conclusions, I would include it at the end of the first paragraph of your Discussion. Without that context, people might conclude that people with AHD have good survival and there's really no reason to perform additional interventions/dedicate additional resources here. Which I believe is incorrect.

Minor point:

The Tables appear to be mislabeled. Two tables are labeled as 'Table 2'.

6. PLOS authors have the option to publish the peer review history of their article (what does this mean?). If published, this will include your full peer review and any attached files.

Reviewer #1: No

Reviewer #2: No

---

## [Author Response · Author response to Decision Letter 0]

7 Nov 2023

Dear Dr. Zijenah, reviewers, 

Thank you for reviewing our manuscript. We hereby submit our point-by-point rebuttal to your comments. The line numbers in our responses reflect those in the tracked version. We think your suggestions have led to an improved manuscript and we hope you find the revised version suitable for publication. 

On behalf of all authors, 

Tinne Gils

Researcher HIV&TB 

Department of Clinical Sciences

Tel: +32/490399978

tgils@itg.be / www.itg.be

Response: We have added the DOI with access to the dataset: https://figshare.com/articles/dataset/dataset_plhiv/24434668. 

Response: We have revised and corrected the reference list. 

Additional Editor Comments:

Under "Data Availability" the Authors stated "No - some restrictions will apply". Authors please elaborate the restrictions.

Response: No restrictions apply. We have corrected the statement. 

Reviewer #1: This feasibility study on the implementation of Advanced HIV disease care package on 1392 PLHIV was very well conducted. The outcome of this study will be useful for the country programs when AHD is getting implemented.

Reviewer #2: In this implementation science study, Gils et al implement the AHD package of care into a TB case finding setting.

Overall this study is well conducted and well explained.

1. I would recommend the authors add further detail regarding the characteristics of the clinic and laboratories that wer e included in this study.

Study setting is only briefly mentioned in lines 93-94.

Please clarify what a "container-based clinic" refers to. And what is meant by a 'mobile unit' (line 94).

What types of clinics were these. On average how many patients are seen per day/week/year. What type of laboratory did they have. How many lab staff were available, and what other types of tests were being performed at the same time? This will help us to better understand implementation of the AHD package of care in context.

Are these busy, high volume clinics/labs that run many other tests? Or are these rural, low volume sites where they have a dedicated lab worker to run the Visitect test?

Response: The clinics were specifically placed for the conduct of the TB TRIAGE+ trial, and all interventions described in the study procedures were performed there. We have extensively adapted the setting and intervention to answer to your questions. Additional information is provided in lines 98-103 and 131-138. 

2. I strongly suspect that you have inaccurate Visitect results (meaning that there are many people with CD4>200 who were misclassified as having a CD4<200), as the authors have suggested in the Discussion. Which would explain high survival (low mortality) among the population with AHD.

As such, I would recommend softening your conclusions (in the paper and in the abstract), accounting for this critical limitation. Specifically in lines 47-48: "AHD nor TB were associated with survival suggesting adequate management referral". I think the lack of association with reduced survival suggests false positive Visitect results contaminating your AHD population. I would add that to your abstract. I find the wording of lines 47-48 to be confusing, with the word "nor" but "neither".

I think the issue of inaccurate Visitect results is so important in accounting for any conclusions, I would include it at the end of the first paragraph of your Discussion. Without that context, people might conclude that people with AHD have good survival and there's really no reason to perform additional interventions/dedicate additional resources here. Which I believe is incorrect.

Response: Indeed, after consultation with the co-authors we agree with the importance of the low accuracy of VISITECT, and we have reflected this in the abstract (lines 50-51), discussion (lines 376-377 and lines 482-485), and conclusion (lines 507-8). 

Minor point:

The Tables appear to be mislabeled. Two tables are labeled as 'Table 2'.

Response: This has been corrected.

---

## [Editor Report · Decision Letter 1]

8 Dec 2023

Implementation of the advanced HIV disease care package with point-of-care CD4 testing during tuberculosis case finding: a mixed-methods evaluation.

PONE-D-23-29770R1

Dear Dr. Gils Tinne,

We’re pleased to inform you that your manuscript has been judged scientifically suitable for publication and will be formally accepted for publication once it meets all outstanding technical requirements.

Kind regards,

Lynn S Zijenah, PhD

Academic Editor

PLOS ONE
---

## [Editor Report · Acceptance letter]

14 Dec 2023

PONE-D-23-29770R1 

PLOS ONE

Dear Dr. Gils, 

I'm pleased to inform you that your manuscript has been deemed suitable for publication in PLOS ONE. Congratulations! Your manuscript is now being handed over to our production team.

Kind regards, 

on behalf of

Professor Lynn S Zijenah 

Academic Editor

PLOS ONE